# Protein *C*-Mannosylation and *C*-Mannosyl Tryptophan in Chemical Biology and Medicine

**DOI:** 10.3390/molecules26175258

**Published:** 2021-08-30

**Authors:** Shiho Minakata, Shino Manabe, Yoko Inai, Midori Ikezaki, Kazuchika Nishitsuji, Yukishige Ito, Yoshito Ihara

**Affiliations:** 1Department of Biochemistry, Wakayama Medical University, 811-1 Kimiidera, Wakayama, Wakayama 641-0012, Japan; shiho@wakayama-med.ac.jp (S.M.); yinai@wakayama-med.ac.jp (Y.I.); ikezaki@wakayama-med.ac.jp (M.I.); nishit@wakayama-med.ac.jp (K.N.); 2Pharmaceutical Department, The Institute of Medicinal Chemistry, Hoshi University, 2-4-41 Ebara, Shinagawa, Tokyo 142-8501, Japan; shino.manabe.e1@tohoku.ac.jp; 3Research Center for Pharmaceutical Development, Graduate School of Pharmaceutical Science & Faculty of Pharmaceutical Sciences, Tohoku University, 6-3 Aoba, Sendai, Miyagi 980-8578, Japan; 4Department of Chemistry, Graduate School of Science, Osaka University, 1-1 Machikaneyama, Toyonaka, Osaka 560-0043, Japan; yukito@chem.sci.osaka-u.ac.jp; 5RIKEN Cluster for Pioneering Research, 2-1 Hirosawa, Wako, Saitama 351-0198, Japan

**Keywords:** *C*-mannosylation, *C*-mannosyl tryptophan, DPY19, thrombospondin type I repeat, cytokine receptor type I

## Abstract

*C*-Mannosylation is a post-translational modification of proteins in the endoplasmic reticulum. Monomeric α-mannose is attached to specific Trp residues at the first Trp in the Trp-x-x-Trp/Cys (W-x-x-W/C) motif of substrate proteins, by the action of *C*-mannosyltransferases, *DPY19*-related gene products. The acceptor substrate proteins are included in the thrombospondin type I repeat (TSR) superfamily, cytokine receptor type I family, and others. Previous studies demonstrated that *C*-mannosylation plays critical roles in the folding, sorting, and/or secretion of substrate proteins. A *C*-mannosylation-defective gene mutation was identified in humans as the disease-associated variant affecting a *C*-mannosylation motif of W-x-x-W of ADAMTSL1, which suggests the involvement of defects in protein *C*-mannosylation in human diseases such as developmental glaucoma, myopia, and/or retinal defects. On the other hand, monomeric *C*-mannosyl Trp (*C*-Man-Trp), a deduced degradation product of *C*-mannosylated proteins, occurs in cells and extracellular fluids. Several studies showed that the level of *C*-Man-Trp is upregulated in blood of patients with renal dysfunction, suggesting that the metabolism of *C*-Man-Trp may be involved in human kidney diseases. Together, protein *C*-mannosylation is considered to play important roles in the biosynthesis and functions of substrate proteins, and the altered regulation of protein *C*-manosylation may be involved in the pathophysiology of human diseases. In this review, we consider the biochemical and biomedical knowledge of protein *C*-mannosylation and *C*-Man-Trp, and introduce recent studies concerning their significance in biology and medicine.

## 1. Introduction

*C*-Mannosylation is a post-translational modification of Trp residues in secretory or membrane proteins [1,2]. *C*-Mannosyl Trp (*C*-Man-Trp), a characteristic structure in *C*-mannosylation, was first identified in ribonuclease 2 (RNase 2) from human urine [3], although hexosylated Trp had already been identified in neuropeptides from the stick insect *Carausius morosus* [4]. Monomeric *C*-Man-Trp was also detected in human urine and blood [5], and the detailed structure was confirmed [6]. The structure of *C*-Man-Trp is unique in the sugar–amino acid linkage because of the C–C bond between mannose and Trp, compared with the C–O bond and C–N bond in *O*-glycans and *N*-glycans, respectively. In the biosynthesis of the *C*-mannosyl Trp structure, a monomeric α-mannose is attached to the first Trp in the Trp-x-x-Trp/Cys (W-x-x-W/C) motif of substrate proteins by the action of *C*-mannosyltransferases in the endoplasmic reticulum (ER) [1,2,7]. A variety of substrate proteins for *C*-mannosylation have been identified in the thrombospondin type I repeat (TSR) and cytokine receptor type I families, suggesting that *C*-mannosylation regulates protein folding and secretion, and thereby affects the functions of parental proteins [2,7]. In mammals, biosynthesis of *C*-mannosylated proteins is done by DPY19L1 and DPY19L3; however, biological functions of protein *C*-mannosylation and metabolism of monomeric *C*-Man-Trp are not yet fully understood. This review describes the chemical and biochemical knowledge of protein *C*-mannosylation and *C*-Man-Trp and the recent progress of studies to reveal their significance in biomedical fields.

## 2. Structural and Synthetic Biology of Protein *C*-Mannosylation and *C*-Man-Trp

### 2.1. The Structure of C-Mannosylated Protein

*C*-Mannosylation is a unique glycosylation in which an α-mannose is directly conjugated with the indole C2 atom of Trp via a C–C bond [3,8] (Figure 1). Protein *C*-mannosylation was firstly found in human RNase 2 [3].

The X-ray structures of a number of the substrate proteins for *C*-mannosylation have been determined. The high-resolution crystal structure of TSR domains of thrombospondin-1 (TSP-1) was initially resolved in 2002 [9]. TSP-1 contains three TSRs; however, only residues from TSR2 and TSR3 were seen in electron density maps. The expression system used in this article does not allow for post-translational *C*-mannosylation. Therefore, no electron density of the mannose linked to Trp was observed. This crystal structure of TSRs revealed that three strands consisted of alternating stacked layers of Trp residues (W-x-x-W-x-x-W motif from A strand) and Arg residues (R-x-R-motif from B strand), capped by disulfide bonds at both ends (Figure 2).

Other crystal structures of the TSR superfamily, such as complements C6 [10], C6 complexed with C5b [11,12], C8 [13], C9 [14], ADAMTS13 [15], MIC2 [16], and properdin [17,18], as well as those of the cytokine receptor type I family, erythropoietin receptor (EPOR) complexed with erythropoietin (EPO) [19], interleukin-21 receptor (IL-21R) bound to interleukin-21 (IL-21) [20], IL-2 receptor (IL-2R) complexed with an IL-2 variant [21], and granulocyte colony-stimulating factor receptor (G-CSFR) complexed with granulocyte colony-stimulating factor (G-CSF) [22], have been resolved. Furthermore, crystal structures of RNase 2 [23], and myelin-associated glycoprotein (MAG) [24] have also been determined. All structures of the TSR superfamily and cytokine receptor type I family described above were stabilized by π-cation interactions between an aromatic ring and a positively charged side chain group [25] of the “Trp-Arg ladder” motif, in which two or three Trp residues (W-x-x-W-x-x-W-x-x-C motif) from one strand were stacked with Arg residues (R-x-R motif) from a second strand. The structures of properdin [18] and IL-21R [20] indicated further stabilization by hydrogen bonds between the mannose and the Arg. NMR structures of TSR domains of F-spondin [26] also adopted a “Trp-Arg ladder” conformation. However, this “Trp-Arg ladder” conformation could not be observed in RNase 2 or MAG.

In several X-ray structures of *C*-mannosylated proteins, the density of the α-mannose linked to Trp fitted a ^1^*C*_4_ chair conformation better than ^4^*C*_1_ conformation [16,18]. However, the electron density of the Trp-linked mannose is not sufficient to clarify the conformation [28].

Molecular dynamics (MD) simulations of the thermal denaturation and reducing conditions of *C*-mannosylated and non-mannosylated TSR2 domains of netrin receptor UNC-5 indicated that the “Trp-Arg ladder” structure in the *C*-mannosylated TSR2 was more rigid compared with the non-mannosylated TSR2. According to MD simulation, the calculated π-cation interactions between the first Trp and Arg in *C*-mannosylated TSR2 disappeared because the first Trp of the W-x-x-W-x-x-W-x-x-C motif was highly flexible and out of the ladder in non-mannosylated TSR2 [29]. In fact, *C*-mannosylated TSR2 showed a higher melting temperature and slower reductive unfolding than non-mannosylated TSR2, revealing that *C*-mannosylation increases the resistance of TSR to thermal and reductive denaturation [29]. In addition, EPOR, RNase 2, and IL-12B with *C*-mannosylated Trp were reportedly more resistant to thermal denaturation than their non-mannosylated counterparts [30]. These results suggest that *C*-mannosylation plays an important role in the structural stability and folding process in substrate proteins.

### 2.2. Synthetic Chemistry of C-Man Trp and the Related Molecules

Since *C*-Man-Trp is not available in sufficient quantities from natural resources, the chemical synthesis of *C*-Man-Trp and *C*-mannosylated peptides has been a powerful tool for biological investigation. Despite the fact that *O*-glycosylation methodology has been developed over the last two decades, *C*-glycoside, especially aryl *C*-glycoside synthetic methodology, is still under development [31,32].

As shown in Scheme 1, Manabe and Ito synthesized *C*-Man-Trp through a nucleophilic attack reaction between lithiated tryptophan derivative **5** and 1,2-anhydro mannose **3** in the presence of BF_3_·OEt_2_ [33]. After the connection of mannose and Trp, protection group manipulation and oxidation, and final deprotection, *C*-Man-Trp was successfully obtained (Scheme 1). ^1^H-NMR analysis based on coupling constants revealed that the synthesized *C*-Man-Trp mainly has a ^1^*C*_4_ conformation. Detailed conformational investigation was not available at that time.

Similar to our synthesis, lactone **9** was used as a mannose moiety precursor instead of 1,2-anhydro mannose **3** (Scheme 2). To the resulting mannoside-carrying indole [34], oxime protected amino acid moiety **12** was introduced. Although the oxime was reduced by Al-Hg in a stereo-random manner to give *C*-Man-Trp and its diastereomer, these stereoisomers were separated.

Nishikawa commenced *C*-Man-Trp synthesis by their original α-selective *C*-alkynylation (Scheme 3) [35]. An indole group was synthesized in the Sonogashira reaction [36] and subsequent Castro reaction [37]. Then, aziridine **18** derived from serine was added to the indole in the presence of Sc(ClO_4_)_3_ to give protected *C*-Man-Trp.

Recently, a Pd(OAc)_2_ catalyzed C–H activation strategy for *C*-Man-Trp synthesis was reported (Scheme 4) [38]. The amide group and quinoline worked as bidentate ligands to the Pd catalyst. After connection of mannose and Trp, the quinoline group was removed by Zn reduction. Similarly, Ni-catalyzed cross-coupling conditions that utilize a Hantzsch ester LED-photoreductant to couple glycosyl bromide and 2-bromo-tryptophan were reported (Scheme 5) [39]. These syntheses are straightforward and the overall yields were high. In addition, photoreductive cross-coupling reactions enable chemical “post-translational” glycopeptide synthesis (Scheme 6). Namely, 2-bromo-tryptophan-containing peptides were prepared by conventional peptide synthesis. Then, Ni-catalyzed cross-coupling reaction with glycosyl bromide gave various glycopeptides. It is noteworthy that strong acidic condition for final deprotection of peptide synthesis is not suitable for *C*-Man-Trp-containing glycopeptides (Scheme 7). The *C*-Man-Trp is anomerized to β-isomer from α-isomer under acidic conditions. The protecting groups removed under basic conditions were chosen. These syntheses dramatically improved synthesis of *C*-glycopeptide synthesis. Organometallic reaction between glycosyl halides and aryl compounds would be powerful strategy for *C*-glycoside synthesis in the near future.

The pyranoside conformation is described by 38 canonical states according to the Cremer–Pople parameter (Figure 3) [40]. Conformation analysis of a mannoside in a protein by *X*-ray crystallography is difficult because the electron density of the mannose is not sufficient and rather flexible. NMR analyses of dynamics and conformation analysis of a mannopyranoside covalently bound to a protein has been reported. For conformational analyses, chemically synthesized peptides and expressed proteins were utilized.

*C*-Mannosylated peptide synthesis is technically limited because *C*-Man-Trp is unstable under acidic conditions for Boc and final deprotection of side chains of amino acids [41]. In order to avoid acidic conditions, an azide group for the amino protecting group and benzyl ether for hydroxy group protection were employed [42]. The azide group is the sterically minimal protecting group and the amino group was regenerated by reduction under mild conditions. These protection strategies offer a minimum protection pattern and allow removal under neutral conditions. Hinou [43] applied our strategy to microwave-assisted solid-phase peptide synthesis. NMR analyses using synthetic 11mer peptides, the W-S-x-W-S motif in EPOR with or without mannoses, showed that *C*-mannosylation stabilized the W-S-x-W-S motif and its surrounding region.

Besides chemical synthesis, mannosylated proteins were reportedly expressed in *Drosophila* Schneider 2 (S2) cells [44]. Mutation of mannose phosphate isomerase (MPI) by CRISPR-CAS can abolish mannose generation activity from glucose (Figure 4). Instead, exogeneous or salvaged mannose was used. This strategy would be applicable to selective ^13^C-labeled mannose incorporation other than *C*-mannosylation. By the addition of ^13^C-labeled mannose, uniformly ^13^C-labeled mannosylated netrin receptor UNC-5 was expressed. Selective ^13^C-labeling enables investigation of mannose dynamics by heteronuclear ^13^C-relaxation experiments, cross-correlated relaxation, and ^13^C-resolved NOESY experiments. ^1^H-NMR coupling constants, cross-correlated relaxation, and NOE data showed *C*-mannose puckers, mainly in ^1^*C*_4_, allowing interconversion between ^1^*C*_4_ and *B*_03_/^1^*S*_3_ states. In addition, *C*-mannose moiety attached to the Trp significantly reduced pyranoside dynamics compared with free mannose.

NMR analyses and MD simulations showed that mannoside in RNase 2 also mainly adopts a ^1^*C*_4_ conformation with ^1^*S*_3_ and ^1^*S*_5_ [28]. These results are consistent with the case of UNC-5. In these examples, besides the *C*-Man-Trp containing peptide, effects of mannosylation on native and denatured conformations were also investigated. The orientation of the mannose in native RNase 2 was different from that in the denatured form. RNase 2 was markedly stabilized only when the α-mannose was bound in the ^1^*C*_4_-synH structure. On the other hand, NMR and MD simulations have shown that ^1^*C*_4_-antiH is preferred in unfolded RNase 2. These results raised an interesting question of how the conformational change occurred. In addition, the presence of *C*-mannose was found to stabilize the conformation of the large insertion loop and *N*-terminal loop through a specific network of hydrogen bonds. It is encouraging that high-level MD simulations provide valuable insights into glycoprotein dynamics that are sufficiently reliable and currently unobtainable by other experimental methods.

## 3. Biochemical Aspects of Protein *C*-Mannosylation and *C*-Man-Trp

### 3.1. Detection and Identification of C-Mannosylation in Proteins

Protein *C*-mannosylation was firstly found and identified in human RNase 2 by MS and NMR [3]. Mass spectrometric analysis of a peptide containing glycosylated Trp by Edman degradation showed a molecular mass 162 atomic mass units higher than that expected for Trp, which suggests that Trp was modified by a hexosyl residue. The product ion spectrum by low-energy collision-induced dissociation (CID) showed a loss of 120 atomic mass units, which correspond to the cleavage of a C_4_H_8_O_4_ moiety. ^1^H-NMR spectrum revealed the absence of indole H2, which indicated that a hexose was directly conjugated with the indole C2 of Trp via a C–C bond. The vicinal ^1^H-^1^H coupling constants and rotating-frame nuclear Overhauser effect spectroscopy (ROESY) intensities indicated that the carbohydrate moiety was an α-mannopyranose [45].

Since the discovery of *C*-mannosylation in RNase 2 from human urine, more than 40 substrate proteins for *C*-mannosylation have been reported (Table 1). The targets of *C*-mannosylation are mostly secretory or membrane proteins, which suggests that *C*-mannosylation is a conventional post-translational modification in the secretory pathway. The substrate proteins of *C*-mannosylation are categorized into three groups: the TSR superfamily, the cytokine receptor type I family, and others.

### 3.2. Analysis and Production of Monomeric C-Man-Trp

A monomeric form of *C*-Man-Trp was initially identified in human urine [5]. In that report, fast atom bombardment mass spectrometry (FAB-MS) exhibited an ion peak at *m/z* values of 367 [M + H]^+^ and 365 [M − H]^−^. FAB-MS, ^1^H-NMR, and ^13^C-NMR spectra suggested that the molecule was a condensation compound of Trp with hexose. The precise structure was later confirmed [6], describing the identification and conformational analysis of the Trp glycoconjugate by HPLC-MS/MS and ^1^H-NMR and ^13^C-NMR. The product ion spectrum of *C*-Man-Trp by CID of the protonated molecule (*m/z* value of 367 [M + H]^+^) showed a fragmentation pattern dominated by the loss of 120 atomic mass units [6], corresponding to the cleavage characteristic of *C*-glycosyl compounds [3]. In addition, the loss of 18 atomic mass units occurred, which corresponds to a loss of a water molecule by sequential retro-aldol cleavage. The product ion spectrum of *C*-Man-Trp by low-energy CID of the deprotonated molecule (*m/z* value of 365 [M − H]^−^) also showed the characteristic loss of 120 atomic mass units [6].

There are some similar hexosylated Trp structures, *C*-Man-Trp, *C*-glucosyl tryptophan (*C*-GlC-Trp), and *N*-mannosyl tryptophan (*N*-Man-Trp) (Figure 1). *C*-GlC-Trp is a *C*-glycosylated amino acid in which an epimer of an α-mannose, α-glucose, is linked to the indole C2 carbon atom of the Trp residue. *C*-GlC-Trp was synthesized and identified by MS/MS, ^1^H-NMR, and ^13^C-NMR [6]; however, *C*-GlC-Trp has yet to be confirmed in nature. In *N*-Man-Trp, α-mannose is connected to the N1 atom of the indole ring of Trp, and *N*-Man Trp was identified in fruits and food samples [6,88], and *Aedes aegypti* chorion peroxidase [89]. Low-energy CID of the protonated and deprotonated molecules (*m/z* values of 367 [M + H]^+^ and 365 [M − H]^−^) of *N*-Man-Trp showed the loss of 162 atomic mass units, corresponding to an intact anhydro-sugar moiety C_6_H_10_O_5_. The product ion spectra of *C*-GlC-Trp were almost identical with that of *C*-Man-Trp [6]. In conclusion, MS/MS experiments could be used to identify the linkage position; however, epimeric mannosyl and glucosyl conjugated Trp could not be differentiated by low-energy CID.

Recently, we established a novel *C*-Man-Trp assay by ultra performance liquid chromatography with hydrophilic interaction liquid chromatography (HILIC-UPLC) [90,91]. We separated similar hexosylated Trp structures of *C*-Man-Trp, *C*-GlC-Trp, and *N*-Man-Trp (Figure 1) and unequivocally identified as different peaks by fluorescence intensities (excitation at 285 nm/emission at 350 nm) or mass abundance (m/z value of 367.15 [M + H]^+^) [91]. By using this assay, we examined tissue/organ distribution of *C*-Man-Trp in mice, and showed that *C*-Man-Trp was abundant in the ovary and uterus [90]. In addition, *C*-Man-Trp had been identified in tissues/organs of ascidians [92] and marine sponges [93].

Although the biosynthetic pathway for *C*-mannosylated proteins has been extensively studied, the mechanism of how monomeric *C*-Man-Trp is produced has not been fully clarified. Recently, we reported that induction of autophagy by nutritional starvation and rapamycin upregulated the level of *C*-Man-Trp in cultured cells, and this increase in *C*-Man-Trp was suppressed in the presence of several inhibitors of lysosomes [91]. Furthermore, chemically synthesized *C*-mannosylated peptides or *C*-mannosylated TSP-1 incubated with lysosomal fractions produced monomeric *C*-Man-Trp. *C*-Man-Trp was not degradable by lysosomes. Thus, autophagic lysosomal degradation at least partly contributed to intracellular *C*-Man-Trp production. Because intracellular *C*-Man-Trp levels may be regulated by other metabolic pathways in additional to the lysosomal pathways, determining the factors that affect intracellular *C*-Man-Trp levels is a forthcoming challenge. It is of interest to clarify the catabolism of *C*-Man-Trp, although *C*-Man-Trp is considered not to be catabolized inside the body, but rather excreted in urine. However, Hossain et al. recently reported that *C*-Man-Trp is further degraded with an enrichment of a bacterial consortium to be utilized as a carbon source [94]. The precise mechanism of *C*-Man-Trp degradation has yet to be clarified, and further elucidation of the catabolic pathway is a future challenge.

### 3.3. Protein C-Mannosylation in Mammals

The deduced biosynthetic pathway of protein *C*-mannosylation is depicted in Figure 5.

#### 3.3.1. The Substrate Proteins for *C*-Mannosylation

Site-directed mutagenesis studies of RNase 2 revealed that *C*-mannosylation was generated at the first Trp in the sequence W_0_-x_+1_-x_+2_-W_+3_ within the protein, and the Trp residue at position +3 could be replaced by Phe. The substitution of Trp at position +3 by Ala abolished the *C*-mannosylation [76]. Prediction of *C*-mannosylation sites by NetCGlyc 1.0 indicated that Trp and Cys residues at position +3 were widely accepted, and that small and/or polar residues such as Ser, Ala, Gly, and Thr residues at position +1 were preferred [95].

#### 3.3.2. *C*-Mannosyltransferase

*Dpy-19* (*dumpy-19*) in *Caenorhabditis elegans* was firstly identified as a gene for *C*-mannosyltransferase [54], and later the homologous genes of *DPY19L1* and *DPY19L3* were also confirmed as the *C*-mannosyltransferase genes in mammals [50,55,62,64].

Niwa et al. identified DPY19L3 as the *C*-mannosyltransferase of R-spondin1 at the first Trp residue in the sequence W-x-x-W [62]. Shcherbakova et al. showed that the first two Trp residues and the third Trp residue in the sequence W-x-x-W-x-x-W-x-x-C of UNC5A were *C*-mannosylated by DPY19L1 and DPY19L3, respectively [55]. Recently, Cirksena et al. examined *C*-mannosylation of the TSP-1 fragment (comprising TSR2 and TSR3 of human TSP-1) in DPY19L1- and DPY19L3-deficient human induced pluripotent stem cells (hiPSCs) [50]. In this study, the first, second, and third Trp residues of W-x-x-W-x-x-W-x-x-C motifs of TSR2 were *C*-mannosylated in wild-type hiPSCs. Neither the first nor second Trp residue was *C*-mannosylated in DPY19L1-deficient cells, and the third Trp residue was not *C*-mannosylated in DPY19L3-deficient cells. Therefore, DPY19L1 is mainly acting on the first two Trp residues in the sequence W-x-x-W-x-x-W-x-x-C of TSP-1, whereas DPY19L3 is acting on the third Trp residue in the consensus sequence, and these results are consistent with a previous report [55]. Meanwhile, Morishita et al. showed that RPE-spondin had two *C*-mannosyl Trp residues in the sequence W-x-x-W-x-x-C [64]. In that study, although the second Trp residue was *C*-mannosylated by DPY19L3, the first Trp was not *C*-mannosylated by DPY19L1-L4. These results suggested the involvement of other unidentified *C*-mannosyltransferase in the first Trp of RPE-spondin. In mouse protein Aster-B (Gramd1b), *C*-mannosylation occurred in non-canonical WAQL [30], although further investigation is still required to identify the *C*-mannosyltransferase in non-canonical sites.

A topological structure of human DPY19L3 was proposed by in silico analysis. DPY19L3 reportedly comprised 11 transmembrane domains and two re-entrant loops with the N- and C-terminal ends facing the cytoplasm and ER lumen, respectively. It was also indicated that the C-terminal region of DPY19L3 was important for the *C*-mannosyltransferase activity [96]. In contrast, it was shown that human DPY19L1 had 13 transmembrane domains, and that some amino acid residues (R211, E220, and R471) on the ER lumen site were essential for *C*-mannosylation and conserved between the DPY-19 family and oligosaccharyltransferase (STT3) family [97]. Structural studies of *C*-mannosyltransferase are required for advanced understanding of protein *C*-mannosylation mechanisms.

#### 3.3.3. Mannose Donors for *C*-Mannosylation

Hofsteenge and colleagues first discovered the biosynthetic pathway of *C*-mannosylated proteins. Protein *C*-mannosylation is enzymatically generated at the first Trp in the consensus amino acid sequence of W-x-x-W in the ER and uses dolichyl-phosphate mannose (Dol-P-Man) as the mannose donor [75]. Dol-P-Man is synthesized from Dol-P and guanosine diphosphate-mannose (GDP-Man) [98,99,100] (Figure 5). Recently, uridine diphosphate-mannose (UDP-Man) was identified in mammals; however, it is not clear whether UDP-Man could be involved in *C*-mannosylation [101].

#### 3.3.4. Functions of Protein *C*-Mannosylation

The functions of the W-x-x-W motif have been studied extensively using synthetic peptides. In the case of TSP-1, peptides containing the W-x-x-W motif bound to heparin and sulfated glycosaminoglycans [102,103]. The W-x-x-W motif in TSP-1 also plays an important role in the maturation of TGF-β through binding to the latent form of TGF-β [104,105]. Furthermore, peptides containing the W-x-x-W motif can exert various biological effects related to angiogenesis [106], T-cell signaling [107], neurite outgrowth [108], and apoptosis [109]. These results suggested that the W-x-x-W motif could have a role in the intermolecular interactions of various functional proteins. However, the effect of *C*-mannosylation on the biological functions of the motif was not elucidated because previous results were obtained simply using synthetic peptides not involved in *C*-mannosylation.

Mutation analysis of the W-S-x-W-S motif of membrane-bound form EPOR indicated that the W-S-x-W-S motif was critical for the cell surface expression of EPOR and ligand-binding [110,111]. However, *C*-mannosylation was not referred to in that study. Later, Furmanek et al. reported that the soluble extracellular domain of EPOR was *C*-mannosylated at the first Trp in the motif, but *C*-mannosylation did not play a role in the secretory process [70]. It is not clear whether this conclusion can be applied to the membrane-bound EPOR.

Similarly to EPOR, the W-x-x-W motif of IL-21R was important for intracellular transport [71]. When wild-type IL-21R and mutants (W195A and R182A) were expressed in HEK293 cells having *C*-mannosyltransferase activity, the wild-type IL-21R was localized in the Golgi apparatus and on the borders of cells. In contrast, W195A, a *C*-mannosylation-defective mutant, and R182A, a W-R-W ladder-defective mutant, were localized within cells, but not on the borders of cells. Moreover, the extracellular domain of IL-21R was secreted out of the cells, but W195A and R182A mutants were not. These results suggested that the interaction between R182 and W195 in the W-R-W ladder may help folding, processing, and transport of IL-21R. In IL-21R, W195 is *C*-mannosylated and takes part in formation of the sugar bridge, whereas the W195 mutation may prevent its formation. Thus, stabilization by the W-R-W ladder and hydrogen bonds between mannose and Arg may be required for the intracellular transport of IL-21R.

In recent years, many studies investigated the function of *C*-mannosylation by using *C*-mannosyltransferase-deficient cell lines and *C*-mannosylation-defective mutants. Several reports demonstrated that *C*-mannosylation is required for proper folding and intracellular transport. When TSR domains of UNC5A or ADAMTS16 were expressed in CHO wild-type cells, DPY19L1- and DPY19L3-deficient cells, ADAMTS16 secretion was reduced in DPY19L1- or DPY19L3-deficient cells [50], and the secretion of UNC5A was reduced only in DPY19L1-deficient cells [55]. The vast majority of the secreted UNC5A from DPY19L1-deficietnt cells was non-*C*-mannosylated and a small portion of that was *C*-mannosylated. All three Trp residues of the W-x-x-W-x-x-W-x-x-C motif of secreted ADAMTS16 were *C*-mannosylated. *C*-Mannosylation of the first two Trp and the third Trp was not detected in the DPY19L1- and DPY19L3-deficient cells, respectively. Intriguingly, non-*C*-mannosylated TSR1 of ADAMTS16 was not secreted. These results suggest that *C*-mannosylation is essential for secretion of ADAMTS16, although *C*-mannosylation is not always required for secretion of UNC5A [50,55]. The secretion levels of ADAMTS16 and UNC5A were reduced in *C*-mannosyltransferase-deficient cells, suggesting that secretion depends on *C*-mannosylation. In addition to UNC5A and ADAMTS16, *C*-mannosyltransferase-deficient cell lines and experimental substitution of Trp for non-*C*-mannosylated amino acids caused disturbance in the secretion of proteins, such as mucins (MUC5AC and MUC5B) [78,79], TSP-1 [50], mindin (Spondin2) [52], ADAMTSL1 [53], MIG-21 [54], ADAMTS13 [56], MIC2 [60], R-spondin1 [62], R-spondin3 [63], ADAMTS4 [67], Isthmin-1 [68], lipoprotein lipase [86], and microfibril-associated glycoprotein 4 (MFAP4) [87]. *C*-Mannosylation also regulated cell surface localization of IL-21R [71] and thrombopoietin receptor (TPOR) [72].

Secretion of *C*-mannosylation-defective mutants of R-spondin2 was reduced in human fibrosarcoma HT1080, whereas its secretion was increased in other human tumor cell lines, such as A549, PANC1, and MDA-MB-231 [65]. Thus, the secretion of R-spondin2 was different among tumor cell lines, indicating that the effect of *C*-mannosylation on R-spondin2 secretion may differ in a cell type-dependent manner. On the other hand, *C*-mannosylation did not affect the secretion of the Ebola virus soluble glycoprotein sGP [82]. Alternatively, secreted hyaluronidase 1 (HYAL1) was not *C*-mannosylated and *C*-mannosylation was considered to negatively regulate its secretion [85]. A computer simulation demonstrated that *C*-mannosylation might cause instability of the HYAL1 conformation [85]. In most substrate proteins for *C*-mannosylation, *C*-mannosylation promotes proper folding and export, but the regulatory role of *C*-mannosylation may differ in each substrate protein.

*C*-Mannosylation was also reported to affect intermolecular interactions of mindin [30,51] and MAG [24]. Mindin binds lipopolysaccharide (LPS) from Gram-negative bacteria. ELISA analysis showed that *C*-mannosylated mindin expressed by HEK293 cells bound to *Salmonella typhosa* LPS, whereas bacterially expressed mindin did not [51]. Conversely, *C*-mannosylated mindin showed a lower affinity for *Escherichia coli* LPS in comparison with unglycosylated mindin by ELISA analysis [30]. Although the reason for these conflicting results remains unclear, *C*-mannosylation may modulate molecular interaction between the substrate protein and its ligands. In the case of MAG, W22 in the W-x-x-W motif (W22 is the first Trp) was *C*-mannosylated, and the *C*-mannosylation at W22 did not occur when the second Trp in W-x-x-W was substituted for Gln (W25Q) [24]. The wild-type and W25Q mutant interacted with the ganglioside GT1b, and W25Q mutant appeared to have a higher affinity for the GT1b compared with the wild-type, which suggests that *C*-mannosylation of MAG may play a regulatory role in the interaction of MAG with gangliosides.

*C*-Mannosylation of R-spondin1, 2, and 3 enhanced Wnt/β-catenin activation [62,63,65]. *C*-Mannosylation was also involved in the regulation of JAK-STAT signaling through effects on TPOR [72] or G-CSFR [73]. *C*-Mannosylation was reported to promote enzymatic activities of ADAMTS4 [67] and lipoprotein lipase [86], whereas *C*-mannosylation of HYAL1 suppressed its enzyme activity [85]. These results suggested that *C*-mannosylation could modulate various functions of enzymes in cells. In addition, *C*-mannosylation of Isthmin-1 affected its *N*-glycosylation [68]. *N*-Glycosylation of Isthmin-1 occurred in *C*-mannosylation-defective mutant cells, although wild-type Isthmin-1 was not *N*-glycosylated.

Several studies reported the effect of *C*-mannosylation on protein folding and stability. *C*-Mannosylation decreased flexibility and promoted the oxidative folding of UNC-5 TSR [29], and was also involved in the redox-dependent folding of mindin [52]. *C*-Mannosylation increased resistance of UNC-5 [29], EPOR [30], RNase 2 [30], and IL-12B [30] to thermal denaturation compared with their non-*C*-mannosylated counterparts.

In summary, *C*-mannosylation plays various roles in intercellular transport, modulating molecular interaction, and structural stability; however, these functions may depend on the types of cells and *C*-mannosylated acceptor substrate proteins.

### 3.4. Protein C-Mannosylation in Non-Mammals

*C*-Mannosyltransferase activity was found in *C. elegans*, amphibians, and birds in addition to mammals, but not in *E. coli* or yeast [41,74]. Recently, it was reported that the predicted topology of *C. elegans* DPY-19 (*Ce*DPY-19), a *C*-mannosyltransferase, comprised 13 transmembrane domains and that mutation of E579 led to almost complete loss of *C*-mannosylation activity [30]. These results suggest that E579 on the ER lumen site is essential for *C*-mannosylation and may play a direct role in substrate recognition and/or catalysis. A yeast microsomal-based in vitro radio-assay using *Ce*DPY-19 and peptides containing the W-x-x-W motif showed that *C*-mannosylation occurred in WSEW, WSEF, and WSEA. Mutation of the WSEW sequence of IL-12B indicated that WSEF and WSEY were *C*-mannosylated at significantly lower modification rates than WSEW [30], which suggests that Phe and Tyr residues at position +3 of the W-x-x-W sequence are moderately acceptable in the context of *Ce*DPY-19. Hypertrehalosaemic peptide in the stick insect *C. morosus* was *C*-mannosylated at a non-W-x-x-W site [83]. This non-*C*-mannosylated counterpart was also found, which indicates that *C*-mannosylation at non-canonical sites may occur at a lower rate than at canonical sites.

Although RNase 2 was not *C*-mannosylated in *Spodoptera frugiperda* Sf9 cells or *Drosophila melanogaster* S2 cells [74], hexosylated Trp was found in the stick insect [4] and *C*-mannosylation of recombinant TSR modules was confirmed in Sf9 cells and *Trichoplusia ni* cells (High-Five) [27]. On the other hand, S2 cells did not *C*-mannosylate TSRs [27]. These results show that insects are capable of *C*-mannosylation, and further investigation of *C*-mannosylation in insects is required.

Protein *C*-mannosylation is also detected in apicoplexan parasites, such as *Plasmodium falciparum* and *Toxoplasma gondii* [59,61,112]. *T. gondii* and *P. falciparum* DPY-19 homologs could *C*-mannosylate TSR domains of the micronemal adhesins TRAP/MIC2 family [59]. In vitro, *T. gondii* and *P. falciparum* DPY-19s were active on a W-x-x-W-x-x-C peptide but inactive on a short W-x-x-W peptide. *T. gondii* and *P. falciparum* DPY-19s used Dol-P-Man as the mannose donor, similar to mammalian *C*-mannosyltransferases. It was revealed that *T. gondii dpy-19* gene is important for the growth fitness of *T. gondii* tachyzoites [113]. Recently, protein *C*-mannosylation is reportedly involved in the parasite virulence of *T. gondii* and the attachment to host cells [60]. Deletion of the *T. gondii dpy-19* gene led to weakened parasite adhesion to host cells, and to reduced parasite motility and host cell invasion. In addition, *C*-mannosyltransferase-deficient parasites showed attenuated virulence and induced protective immunity in mice [60]. On the other hand, *P. falciparum dpy-19* gene was dispensable, and considered unlikely to play a critical role in the asexual blood stages of the parasite [114]. Meanwhile, DPY-19 protein is reportedly involved in the establishment of left/right asymmetry in neuroblast migration of *C. elegans* [115]. Collectively, biological roles of *C*-mannosylation may be different in each species of parasite. Although the precise regulatory mechanisms were not fully examined, these studies suggested that protein *C*-mannosylation might exert some certain functions in the lifecycle or pathogenic process of parasites. Thus, these results suggest that protein *C*-mannosylation is involved in the regulation of a variety of biological events in various living organisms.

## 4. Biomedical Topics of Protein *C*-Mannosylation and *C*-Man-Trp

### 4.1. Protein C-Mannosylation in Disease

Defects in *C*-mannosyltransferase and mannose donors, and gene mutations related to the W-x-x-W motif of proteins are considered to be associated with congenital human diseases due to impaired protein *C*-mannosylation. However, there has been no human disease linked to the defect of *DPY19L1* or *DPY19L3* genes.

#### 4.1.1. Diseases by Defects in *C*-Mannosyltransferase

Several phenotypes of the *C*-mannosylatransferase knockout mouse strains are disclosed in the Web databank of the International Mouse Phenotyping Consortium (IMPC) [116]. Knockout of *DPY19L1* gene showed several abnormalities, such as: enlarged heart, small kidney, abnormal seminal vesicles, and decreased fasting glucose levels in blood. In addition, other abnormal phenotypes include preweaning lethality or the absence of an expected number of homozygote pups based on Mendelian ratios. DPY19L1 knockdown was also reported to impair radial neuronal migration in mouse corticogenesis [117]. In *DPY19L3* gene knockout, there were several blood abnormalities leading to reductions in the erythrocyte number, hematocrit, free fatty acid levels, and hemoglobin content (IMPC). It was recently reported that *DPY19L3* gene knockout impairs eye development in the Japanese rice fish medaka (*Oryzias latipes*) via the dysfunction of ADAMTS16 without *C*-mannose [50]. These data suggest that *C*-mannosylation defects in proteins may cause a variety of developmental disorders and diseases through the dysfunction of the substrate proteins. Although it is not clear whether DPY19L2 is a *C*-mannosyltransferase, impairment of the *DPY19L2* gene was reported in patients with globozoospermia, in which sperm development is disturbed [118]. Thus, further investigation is required to elucidate the pathogenic mechanism related to the defect of DPY19L2 in humans.

#### 4.1.2. Diseases by Defects in Mannose Donors

Aberrant metabolism with dolichol and/or mannose is also considered to affect protein *C*-mannosylation, because *C*-mannosyltransferases require Dol-P-Man as a mannose donor for *C*-mannosylation (Figure 5) [75]. There have been several examples reported as congenital metabolic disorders related to Dol-P-Man, and the defective genes include genes for Dol-P synthesis (*NUS1*, *DHDDS*, *SRD5A3*, and *DOLK*), GDP-Man synthesis (*PMM2* and *GMPPB*), and Dol-P-Man synthesis or utilization (*DPM1*, *DPM2*, *DPM3*, and *MPDU1*) [119,120]. Dysfunction of these genes is thought to affect *N*-glycosylation, *O*-mannosylation, *C*-mannosylation, and GPI-anchor biosynthesis in proteins, which are known as congenital disorders of glycosylation (CDG) with heterogenous clinical symptoms with neural, cardiac, or muscular abnormalities [120]. In the case of *DPM3*-gene deficiency, protein *O*-mannosylation is mainly affected, and the loss of α-dystroglycan *O*-mannosylation is considered to cause muscular dystrophy [121]. In that study, *C*-mannosyl structures were slightly affected but not lost in properdin from DPM3-deficient patient plasma, which indicates that the residual Dol-P-Man levels are sufficient for *C*-mannosylation in DPM3-deficient patient. Thus, this study suggested that the *C*-mannosylation reaction is not so sensitive to the availability of Dol-P-Man in the cell, although it has yet to be fully examined how protein *C*-mannosylation is influenced in other CDG-related diseases associated with Dol-P-Man synthesis.

#### 4.1.3. Diseases Related to Protein *C*-Mannosylation

A *C*-mannosylation-defective mutation of *ADAMTSL1* was identified as the first disease-associated variant affecting the *C*-mannosylation of the W-x-x-W motif, which caused various phenotypes including developmental glaucoma, myopia, and retinal defects [122]. Recently, it was revealed that a gene mutation disrupting the W-S-x-W-S motif in IL-2R caused alteration in the IL-2R structure and decreased protein levels, resulting in immune dysregulation related to T and NK cells [123]. Thus, their studies suggested that novel human diseases with gene mutations, in which the W-x-x-W motif for *C*-mannosylation is disrupted, would be identified more in the future.

By using an antibody against *C*-Man-Trp, we previously showed that protein *C*-mannosylation is increased in the aortic tissues of the diabetic Zucker fatty rats [124]. TSP-1 was identified as a target protein for *C*-mannosylation in aortic tissue lysate of the diabetic rats. TSP-1 was reportedly increased in the aortic vessel walls of diabetic Zucker fatty rats, which suggested that TSP-1 is involved in the atherosclerotic change of vascular tissues in the presence of diabetes [125]. Collectively, these results suggest a possible pathological link between upregulated *C*-mannosylated TSP-1 and vascular damage in diabetes.

#### 4.1.4. Research Use of Synthetic Peptides

To investigate how *C*-mannosylation of the W-x-x-W motif involved in human diseases, we took advantage of synthetic approach. Synthetic pathways for *C*-Man-Trp and *C*-mannosylated peptides have been established by Manabe et al. and Nishikawa et al. [33,126], which enables us to investigate the biochemical function of *C*-mannosylated W-x-x-W motifs. By using synthetic peptides with *C*-mannose, we showed that *C*-mannosylated tetrapeptide of *C*-Man-WSPW but not *C*-Man-Trp demonstrated an enhancing effect on the cytotoxicity of LPS by increasing the production of tumor necrosis factor-α (TNF-α), which suggested that the *C*-mannosylated TSR-derived W-x-x-W motif is involved in the regulation of LPS-related innate immunity [127].

We also searched for the target protein bound to *C*-mannosylated WSPW peptide, and found that heat shock cognate protein 70 (Hsc70), a molecular chaperone in the cytosol and nucleus, preferably binds the *C*-mannosylated peptide rather than the non-mannosylated one in macrophage-like RAW264.7 cells [128]. Furthermore, the Hsc70-induced production and secretion of TNF-α were enhanced more by *C*-Man-WSPW than WSPW. These results indicated that the *C*-Man-WSPW peptide exerts a modulatory function in the Hsc70-related signaling pathway, although the mechanical correlation between the enhanced LPS-induced signaling with the *C*-mannosylated peptide and Hsc70 was not clear.

### 4.2. C-Man-Trp in Disease

A monomeric form of *C*-Man-Trp was initially identified in human urine and blood [5]. Takahira et al. were the first to report that the level of *C*-Man-Trp is upregulated in sera of patients with renal dysfunction [129] and indicated that the measurement of *C*-Man-Trp is a better indicator for renal dysfunction than that of serum creatinine, because the level of *C*-Man-Trp is not affected by the muscle mass. Since then, there have been several studies supporting the utility of *C*-Man-Trp as a diagnostic serum marker of renal damage in humans [130,131,132,133,134,135].

Renal dysfunction is a major complication of diabetes in the later phase. Upregulated serum *C*-Man-Trp may be a risk factor or prognostic biomarker for chronic kidney disease in patients with diabetes [131,133,135], but a causal linkage between *C*-Man-Trp and diabetes has not been fully examined. To further investigate the relation of *C*-Man-Trp with diabetes, we examined the blood level of *C*-Man-Trp in the KK-Ay mouse, a model of type 2 diabetes, by using UPLC-MS [90]. We found that the level of *C*-Man-Trp in plasma was unexpectedly comparable between KK-Ay mice with hyperglycemia and controls despite albuminuria, which suggested that the upregulation in plasma *C*-Man-Trp may reflect a greater severity of renal damage in the presence of diabetes. Recently, it was reported that higher urine *C*-Man-Trp levels were associated with a higher risk of adverse kidney events, such as kidney failure and acute kidney injury, and death [136]. We examined the level of serum *C*-Man-Trp in patients with type 2 diabetes, and found that the level of serum *C*-Man-Trp is significantly correlated with that of the creatinine or cystatin C, indicating that serum *C*-Man-Trp is useful to estimate the renal function in patients with type 2 diabetes [137]. It is also noteworthy that the level of serum glucose showed no correlation with that of serum *C*-Man-Trp of diabetic patients whose renal function is maintained (eGFR > 60 mL/min/1.73 m^2^). The results are consistent with other reported results with comparable levels of serum *C*-Man-Trp between KK-Ay diabetic mice and non-diabetic controls [90].

In type 2 diabetic patients, the level of serum *C*-Man-Trp was increased with the progression of peripheral artery disease as assessed by ankle-brachial pressure index (ABI) [137]. This may be compatible with the fact that *C*-mannosylated TSP-1 was upregulated in aortic tissues of diabetic Zucker fatty rats [124]. These results suggest a pathological correlation between protein *C*-mannosylation and monomeric *C*-Man-Trp in vascular complications under diabetic conditions. Furthermore, in a serum metabolic profiling study for human mortality, *C*-Man-Trp showed a significant correlation with cardiovascular disease mortality [138]. Taken together, serum *C*-Man-Trp could be clinically useful for assessing glomerular filtration function and vascular complications, although it is not a simple biomarker for hyperglycemia in diabetes patients.

Based on the characteristic distribution of *C*-Man-Trp in ovarian tissues [90], we focused on the relation between *C*-Man-Trp and ovarian diseases. We found that the *C*-Man-Trp level in plasma was significantly higher in the malignant ovarian tumor than in the borderline and benign tumor groups or the normal control group [139]. The plasma *C*-Man-Trp identified malignant ovarian cancer with high sensitivity, and the combination of *C*-Man-Trp with CA125, an authentic marker of ovarian cancer [140], showed even better diagnostic performance than either marker alone. Thus, measuring blood *C*-Man-Trp, alone or in combination with CA125, may allow for early non-invasive screening of ovarian cancer. Regarding *C*-Man-Trp and ovarian cancer, it is not clear how *C*-Man-Trp is upregulated in the blood of ovarian cancer patients. As described above, *C*-Man-Trp is produced in part by degradation of *C*-mannosylated proteins through the autophagic pathway in cells [91]. On the other hand, it was reported that *C*-mannosylation of R-spondin2 is involved in the promotion of cancer cell migration in various human tumor cells, suggesting a pathological correlation between protein *C*-mannosylation and tumor progression [65]. Thus, it is of interest to investigate the correlation of altered expression of *C*-mannosylated proteins and upregulated production of monomeric *C*-Man-Trp in cancer tissues. Together, these results suggest that measuring blood *C*-Man-Trp could be a reliable approach to ovarian cancer diagnosis, although the renal dysfunction could affect the clinical assessment. Further studies are also needed to assess the utility of *C*-Man-Trp measurement for distinguishing among malignant ovarian tumors of different histological subtypes and stages. Moreover, the clinical assessment of *C*-Man-Trp should be explored in malignancies derived from various different tissues or organs other than the ovaries.

In regard to other medical aspects, the level of *C*-Man-Trp in fasting blood is reportedly correlated with the age and aging traits including the lung function and bone mineral density [141]. Recently, this was supported by the finding that the level of *C*-Man-Trp in plasma is correlated with frailty in metabolomics analyses [142]. The serum level of *C*-Man-Trp was also reported to be correlated with the elevated state of inflammation in older adults [143]. In addition, it was shown that urinary secretion of *C*-Man-Trp increases in patients with irritable bowel syndrome [144]. In that study, degradation of mucosal layer of the intestines and presence of low-level inflammation are thought to be involved in the increased urinary secretion of *C*-Man-Trp, although the precise pathogenic mechanism has yet to be clarified. Recently, it was also reported that serum *C*-Man-Trp was closely associated with pre-term pre-eclampsia at 28 weeks of gestational age [145], although the pathological correlation with *C*-Man-Trp was not clear. Thus, aging and inflammation may also be important factors affecting the level of *C*-Man-Trp in the body, and further exploration is needed to clarify the functions of *C*-mannosylation in human health and diseases.

## 5. Conclusions

Protein *C*-mannosylation plays functional roles in the proper folding and secretion, and is also involved in intermolecular interactions and regulation of receptor-mediated signaling pathways. Several reports demonstrated the correlation between protein *C*-mannosylation and diseases. The knockout of *C*-mannosyltranseferase caused several abnormalities in mice, and defects in eye development in medaka fish. Recently, a disease-associated variant in the *C*-mannosylation motif was firstly identified in humans. Monomeric *C*-Man-Trp is in part produced by lysosomal degradation of *C*-mannosylated proteins in cultured cells. However, intracellular *C*-Man-Trp levels may be also regulated by other metabolic pathways. The level of *C*-Man-Trp in blood is increased in patients with renal dysfunction or ovarian cancer, which suggests that *C*-Man-Trp in blood may be a biomarker to assess various conditions of health and disease in humans. Further investigations concerning the biosynthesis and functions of *C*-mannosylated proteins and *C*-Man-Trp are required to clarify the biological and medical significance of protein *C*-mannosylation and *C*-Man-Trp.

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
