# Peer review of "Protein *C*-Mannosylation and *C*-Mannosyl Tryptophan in Chemical Biology and Medicine"

_molecules, 2021, doi:10.3390/molecules26175258_

Round 1
Reviewer 1 Report
This is a beautifully written and timely review providing a much-needed update on progress in the field of tryptophan C-mannosylation. I thoroughly enjoyed reading this work and would like to congratulate the authors on a job well done. I recommend publication following a few small additions, detailed below.
Section 2.2.
This must be updated to incorporate the recent JACS publication (https://doi.org/10.1021/jacs.1c05567) that describes superior methods for the synthesis of C-Man-Trp and the direct installation of this modification on peptides. This is a very significant advance in the field and should not be omitted.
Table 1
This table is fantastic, though the changes suggested below would make it even better:
1) include the species and accession number for each protein entry
2) state if the sample was obtained from a primary source or via homologous or heterologous over expression
3) add missing entries from ref #30 in the ‘TSR’ family: Hemicentin-1 (Hmcn1); Spondin-1 (Spon1); Semaphorin 5a (Sema5a); Adhesion G protein-coupled receptor B1 (Adgrb1); Thrombospondin type-1 domain-containing protein 7A (Thsd7a); Thrombospondin type-1 domain-containing protein 7B (Thsd7b); and 'other' family: Ribitol-5-phosphate xylosyltransferase 1 (Rxylt1); Protein Aster-B (Gramd1b)
Section 3.2.
I was a little confused by the addition of C-glucosyl Trp to this section. Is there any evidence that this exists in Nature? If not, please state this clearly in this section.
Other.
Some further discussion of C-mannosylation at non-WXXW/C sites is warranted. For example, the mouse Gramd1b protein is modified at ‘WAQL’ and in the stick insect Carausius morosus the Carmo-HrTH-I peptide is modified at ‘WGTG’ (see UniProt A0A6G4ZU62). Both assignments are well supported by tandem MS data. I think these observations tease at interesting future directions for research, namely exploring how DPY19 sequence and structure dictate substrate specificity.
Author Response
Manuscript ID: molecules-1336242
We greatly appreciate your review of our manuscript and the reviewers’ helpful suggestions. Below are our responses to the reviewers’ comments, together with a description of the changes we have made to our manuscript. In the revised manuscript, red text indicates the portions that were revised according to the reviewers’ comments.
Reviewer #1:
This is a beautifully written and timely review providing a much-needed update on progress in the field of tryptophan C-mannosylation. I thoroughly enjoyed reading this work and would like to congratulate the authors on a job well done. I recommend publication following a few small additions, detailed below.
Section 2.2.
This must be updated to incorporate the recent JACS publication (https://doi.org/10.1021/jacs.1c05567) that describes superior methods for the synthesis of C-Man-Trp and the direct installation of this modification on peptides. This is a very significant advance in the field and should not be omitted.
Response- First, we thank the reviewer for the careful and critical reading of our manuscript and for her/his constructive comments. We addressed the latest JACS report on page6, lines152-163, Scheme5-7, and ref #39.
Table 1
This table is fantastic, though the changes suggested below would make it even better:
1) include the species and accession number for each protein entry
2) state if the sample was obtained from a primary source or via homologous or heterologous over expression
3) add missing entries from ref #30 in the ‘TSR’ family: Hemicentin-1 (Hmcn1); Spondin-1 (Spon1); Semaphorin 5a (Sema5a); Adhesion G protein-coupled receptor B1 (Adgrb1); Thrombospondin type-1 domain-containing protein 7A (Thsd7a); Thrombospondin type-1 domain-containing protein 7B (Thsd7b); and 'other' family: Ribitol-5-phosphate xylosyltransferase 1 (Rxylt1); Protein Aster-B (Gramd1b)
Response- Thank you for the good suggestions. Table 1 was modified based on your suggestions.
1) We added columns for “species” and “UniProt accession” to the Table 1.
2) We also added a column for “primary source / recombinant protein”.
3) We added ref #30 in Properdin, F-spondin, RPE-spondin. In addition, we added proteins of Thrombospondin type-1 domain-containing protein 7A (Thsd7a), Thrombospondin type-1 domain-containing protein 7B (Thsd7b), Adhesion G protein-coupled receptor B1 (Adgrb1), Hemicentin-1 (Hmcn1), and Semaphorin 5a (Sema5a) in the TSR family, and Ribitol-5-phosphate xylosyltransferase 1 (Rxylt1) and Protein Aster-B (Gramd1b) in Others.
Section 3.2.
I was a little confused by the addition of C-glucosyl Trp to this section. Is there any evidence that this exists in Nature? If not, please state this clearly in this section.
Response- Thank you for the suggestion. We addressed this point on page 13, lines 258-260.
Other.
Some further discussion of C-mannosylation at non-WXXW/C sites is warranted. For example, the mouse Gramd1b protein is modified at ‘WAQL’ and in the stick insect Carausius morosus the Carmo-HrTH-I peptide is modified at ‘WGTG’ (see UniProt A0A6G4ZU62). Both assignments are well supported by tandem MS data. I think these observations tease at interesting future directions for research, namely exploring how DPY19 sequence and structure dictate substrate specificity.
Response- Thank you for your constructive comments. We discussed about C-mannosylation of Gramd1b at a non-canonical site on page 15, lines 333-335, and that of stick insect HrTH-I peptide on page 17, lines 464-467.
Reviewer 2 Report
Minakata and collaborators present a thorough review of C-mannosylation in biomedical and biotechnology contexts, covering from structural aspects to disease relevance. The work is well organized, written in a comprehensive manner and displays a style that indicates authority in the field.
Suggestions:
- Only minor spell checking required.
Author Response
Manuscript ID: molecules-1336242
We greatly appreciate your review of our manuscript and the reviewers’ helpful suggestions. Below are our responses to the reviewers’ comments, together with a description of the changes we have made to our manuscript. In the revised manuscript, red text indicates the portions that were revised according to the reviewers’ comments.
Reviewer #2:
Minakata and collaborators present a thorough review of C-mannosylation in biomedical and biotechnology contexts, covering from structural aspects to disease relevance. The work is well organized, written in a comprehensive manner and displays a style that indicates authority in the field.
Suggestions:
- Only minor spell checking required.
Response- First of all, we would like to thank the reviewer for having devoted their time to reviewing our manuscript. Following typos are corrected, “Research” (on page 1, line 11), “organometallic” (on page5, line 148), and “Hypertrehalosaemic hormone” (on page 11, Table 1).